# Platelets as Key Factors in Hepatocellular Carcinoma

**DOI:** 10.3390/cancers11071022

**Published:** 2019-07-20

**Authors:** Natasa Pavlovic, Bhavna Rani, Pär Gerwins, Femke Heindryckx

**Affiliations:** 1Department of Medical Cell Biology, Uppsala University, Box 571, Husargatan 3, 75-431 Uppsala, Sweden; 2Department of Radiology, Uppsala University Hospital, Sjukhusvägen 85, 751-85 Uppsala, Sweden

**Keywords:** hepatocellular carcinoma, platelets, hemostasis, fibrosis, tumor-stroma interactions, hepatic stellate cells, macrophages

## Abstract

Hepatocellular carcinoma (HCC) is a primary liver cancer that usually develops in the setting of chronic inflammation and liver damage. The hepatic microenvironment plays a crucial role in the disease development, as players such as hepatic stellate cells, resident liver macrophages (Kupffer cells), endothelial cells, extracellular matrix, and a variety of immune cells interact in highly complex and intertwined signaling pathways. A key factor in these cross-talks are platelets, whose role in cancer has gained growing evidence in recent years. Platelets have been reported to promote HCC cell proliferation and invasion, but their involvement goes beyond the direct effect on tumor cells, as they are known to play a role in pro-fibrinogenic signaling and the hepatic immune response, as well as in mediating interactions between these factors in the stroma. Anti-platelet therapy has been shown to ameliorate liver injury and improve the disease outcome. However, platelets have also been shown to play a crucial role in liver regeneration after organ damage. Therefore, the timing and microenvironmental setting need to be kept in mind when assessing the potential effect and therapeutic value of platelets in the disease progression, while further studies are needed for understanding the role of platelets in patients with HCC.

## 1. Introduction

Hepatocellular carcinoma (HCC) is a primary liver cancer that usually develops in the setting of chronic liver damage. In most cases it is diagnosed at a late stage which limits therapeutic options, making it the third leading cause of cancer-related death worldwide [1]. Hepatocellular carcinoma initiation and progression are set in a background of chronic inflammation, which creates a micro-environment favorable for tumor growth [2]. This environment includes hepatic stellate cells (HSC), resident liver macrophages (Kupffer cells), endothelial cells, extracellular matrix (ECM), a variety of immune cells, as well as newly-formed, leaky and dysfunctional blood vessels that typically occur in cirrhosis and HCC [3,4,5]. These leaky blood vessels allow tumor cells to interact with several components of the hemostatic system, thereby leading to the activation of the coagulation cascade and altering hepatic hemodynamics [6,7]. An aspect of the hemostatic microenvironment that has gained scientific attention is the involvement of platelets in tumor progression and invasion.

Platelets are small, discoid blood elements formed as anuclear cytoplasmic vesicles from megakaryocytes in the bone marrow. Their key role in hemostasis is the initiation of the coagulation cascade in response to vascular injury, where they adhere to the ECM to form a blood clot and become activated by being exposed to collagen [8]. The range of functions that platelets exhibit beyond clotting can be explained by the extensive amount of protein they are able to synthesize, express, and release [9]. Upon activation, platelets release α-granules and dense granules that contain inflammatory cytokines, chemokines, and multiple growth factors such as platelet-derived growth factor (PDGF), serotonin, endothelial growth factor (EGF), insulin-like growth factor 1 (IGF-1) transforming growth factor beta (TGFβ), tumor necrosis factor alpha (TNFα), interleukin-6 (IL-6), chemokine (C-X-C motif) ligand 4 (CXCL4), vascular endothelial growth factor A (VEGF-A), hepatocyte growth factor (HGF), and fibroblast growth factor (FGF) (Figure 1) [10]. The same mediators that promote wound-healing in a normal physiological state could have adverse effects in a tumor microenvironment. This is particularly interesting in HCC; as both liver cirrhosis and cancer are conditions that can perturb the hemostatic balance towards a pro-thrombotic state, thus creating a hypercoagulable condition that further influences tumor cell behavior [11].

Platelets are actively recruited to the liver upon organ damage and are known to play a vital role in tissue regeneration, mainly through secreting high concentrations of serotonin and promoting hepatocyte proliferation [8]. Besides a potential direct effect on HCC cells, platelets interact with different cell types in the stroma, including hepatic stellate cells, endothelial cells, and hepatic immune cells [12,13] (Figure 1). Their involvement in the tumor-stroma interplay has been reported to contribute to a more aggressive and metastatic tumor phenotype in HCC [14,15,16] and other solid tumors [17,18]. Within the HCC microenvironment, platelet-derived factors directly influence tumor cell proliferation, as well as pro-fibrinogenic signaling and immune cell recruitment, while also mediating the cross-talk between these different processes in the stroma. In this review, we provide an overview on the role of platelets in the pathogenesis of HCC, both by focusing on the direct effect on tumor cells and their role in the tumor microenvironment (Figure 1).

## 2. The Effect of Platelets on HCC Proliferation and Metastasis

Several in vitro and in vivo studies have shown that platelets have a strong proliferative effect on hepatocytes. Matsuo et al. found that platelet administration and thrombocytic conditions potently induce liver regeneration after hepatectomy in mice and rats, while in vitro studies showed that direct contact between platelets and hepatocytes was necessary to induce a proliferative effect through VEGF, HGF, and IGF-1 signaling [19,20,21]. Platelets enhance liver regeneration by direct signaling with hepatocytes, as well as through parenchymal cells of the liver, such as Kupffer cells and liver sinusoidal endothelial cells (LSECs). The direct effect platelets exert on hepatocyte proliferation is triggered by secretion of IGF-1, HGF, TGFβ, VEGF, PDGFβ which activates cognate receptors to enable downstream signaling, ultimately resulting in cell cycle progression [20,22,23,24]. Signaling cascades associated with this platelet-mediated effect include TNFα/nuclear factor-kappa B (NF-κB), IL-6/signal transducer and activator of transcription 3 (STAT3), and phosphatidylinositol-3-kinase (PI3K)/Akt. Platelets have also been found to have a strong proliferative effect on LSECs through direct contact, resulting in further hepatocyte proliferation and liver regeneration, mainly through increased secretion of EGF and IL-6 by LSECs, which promotes DNA synthesis in hepatocytes [25,26]. Another study found that platelets significantly amplify leukocyte- and Kupffer cell-dependent hepatocyte proliferation through TNFα and IL-6 secretion [27,28].

Although the involvement of platelets in metastasis has been extensively investigated in other tumors, less is known about the effect of platelets on tumor cell growth and metastasis in HCC [17,29,30]. A clinical study showed that platelet counts were higher in HCC patients with extrahepatic metastases compared to those without metastases, indicating a possible role for platelets in HCC metastasis [31]. One notable way platelets support tumor cell migration is by adhering to them through adhesion receptors GPIIb/IIIa, GPIb-IX-V and P-selectin, thereby protecting them from immuno-surveillance and shear forces in the blood flow, as well as supporting their arrest to the vessel wall [32]. In vivo and in vitro studies on melanoma and breast cancer metastasis have shown that platelets activate the coagulation cascade by secreting thrombin and tissue factor, which results in a meshwork of platelets and fibrin shielding tumor cells and allows them to escape immune-surveillance and successfully invade distant sites [29,33]. A recent paper by Zhuang et al, shows evidence that this process also occurs in a metastatic HCC-mouse model [34]. They found that activated platelets adhere to tumor cells and that pharmacological inhibition of platelet activation, with diosgenin and diosgenin derivates, inhibits platelet adherence to tumor cells and decreases metastasis. In vitro studies have also shown a direct effect of platelet lysates on tumor cell migration and invasion, suggesting the pro-metastatic effect is not merely a result of avoiding immune-surveillance, but also due to the direct effect of their stored growth factors [16,34]. Platelets have also been reported to alter the response to chemotherapeutic agents. Exposure of HCC cell lines to platelet lysates antagonizes the effect of Sorafenib and Regorafenib, thus suggesting platelets could play a role in chemoresistance, probably via the release of EGF and IGF-1 [35]. This has also been shown in other tumors, as adenocarcinoma cells become more resistant to anticancer drugs after exposure to platelets [36] and clinical data from breast cancer patients show that tumor cells surrounded with platelets are less responsive to neo-adjuvant chemotherapy [37].

An important factor in the cross-talk between platelets and tumor cells is serotonin. Tumor growth in HCC is strongly enhanced by serotonin, most of which is circulating in the bloodstream and is transported by platelet dense granules. In vitro experiments have demonstrated that serotonin induces the proliferation of three different HCC cell lines, while the inhibition of serotonin signaling suppressed tumor growth in two tumor mouse models [38,39]. Another in vitro study found that serotonin protects HCC cells from starvation-induced cell death, as well as that cancer cells overexpressed serotonin receptor-2B in a mouse HCC model [40]. A clinical study on HCC patients reported that intra-platelet serotonin levels were positively correlated with tumor growth and cancer progression. Furthermore, this parameter indicated poorer recurrence-free and overall survival [41]. Another important factor that drives platelet-induced tumor cell proliferation, is the release of TGFβ, which is stored in the α-granules. He et al. found that platelet releasates strongly stimulate HCC cell proliferation both in vivo and in vitro by decreasing the expression of Krüppel-like factor 6 (KLF6), a tumor suppressor that has been shown to repress HCC proliferation and metastasis [42]. The study revealed that blocking TGFβ signaling diminished the inhibitory effect of platelets on KLF6 expression, highlighting platelet-derived TGFβ as a key factor in this cross-talk. This effect was also absent after the silencing of KLF6 in HCC cell lines and when HCC cells were incubated with platelets exhausted of their releasates [22]. Platelet-derived TGFβ has been found to induce a TGFβ/Smad and NF-κB—cooperated signaling cascade in tumor cells to promote epithelial-mesenchymal transition into a pro-metastatic phenotype and allow their extravasation and metastasis [22]. In turn, tumor cells have been shown to release adenosine diphosphate (ADP), thrombin, and tissue factor in response to platelet-initiated coagulation steps, which propels the platelet activation and coagulation cascade to further support tumor invasion and metastasis [29].

## 3. Platelets, Liver Sinusoidal Endothelial Cells, and Angiogenesis

The interaction between endothelial cells and platelets plays an important role in mediating liver regeneration after organ damage. It acts as a fibrosis gatekeeper by maintaining a quiescent phenotype of HSC, which reside close to the endothelium, namely in the space of Disse. In vitro studies have shown that platelets promote LSEC proliferation and activation, resulting in VEGF and IL-6 production, as well as that direct contact between LSECs and platelets has a proliferative effect on hepatocytes [26].

The formation of new blood vessels is crucial in the pathogenesis of chronic liver disease, both during fibrosis and in tumorigenesis [3]. In an environment of cirrhosis and HCC, the leaky, structurally and functionally abnormal blood vessels provide a source of oxygen and nutrients, which becomes a potent driver of tumor growth and invasion [43]. Pro-angiogenic factors such as VEGF and FGF are stored in the platelet granules, which contributes to the angiogenic switch, but also directly stimulates tumor cell growth and proliferation. Elevated VEGF levels have been detected in serum and tumor tissue of HCC patients and shown to correlate with more invasive disease and shorter survival [44]. There is also a positive correlation between levels of circulating angiogenic factors and platelet count in HCC patients, which has been associated with poor prognosis [45,46,47,48]. Although cancer cells and other cells from the HCC microenvironment are also rich sources of VEGF, there are indications that platelets play a pivotal role in mediating angiogenesis in HCC. An in vitro study showed that purified platelets, regardless of their activation status and content release, promote tube formation in human umbilical vein endothelial cells [29]. Several studies have confirmed that platelets are an important carrier of circulating angiogenic factors [49]. Platelets could also play a role in regulating angiogenesis by direct contact with endothelial cells, which would further propel malignancy in a tumor setting, considering the large number of leaky vessels present in the tumor micro-environment. During hepatic injury, platelets infiltrate the liver and interact with LSECs, where they become activated in response to damaged endothelium [13]. Platelets and LSECs have close functional interactions which are partly mediated by P-selectin, which is expressed on both activated platelets and endothelial cells [50]. Studies on P-selectin-knockout mice have shown decreased tumor growth and metastasis in colon cancer [51]. Selectin-dependent interactions between platelets, leukocytes, and tumor cells are associated with an increase in endothelial cell activation and chemokine-5 production, which has a pro-metastatic effect through monocyte recruitment [29]. By expressing significantly higher levels of P-selectin than endothelial cells, platelets become crucial factors in promoting neutrophil recruitment, thereby serving as an important mediator between neutrophils and LSECs [25,52]. While neutrophils have a key role in pathogen clearance and host defense in the liver, their excessive activation can have adverse effects in an inflammatory setting. For example, neutrophils can promote hepatic metastasis in vivo by stimulating angiogenesis, mainly via the secretion of FGF-2, which would further fuel tumor growth [53]. Another important aspect of platelet-mediated activation of the endothelium is carried out through ligation of CD40 expressed on endothelial cells by platelet-derived CD40L. In vitro experiments have shown that CD40L, expressed as part of the basic platelet reaction during their activation cascade, induces endothelial cells to produce adhesion molecules E-selectin (CD62E), VCAM-1 (CD106), and ICAM-1 (CD54) and chemokines IL-8 and MCP1, resulting in leukocyte and monocyte recruitment to the site of injury, which could accelerate metastasis [29,54]. Studies have also found that platelet-driven leukocyte recruitment induces liver damage during systemic endotoxemia in rats, as well as that infiltrating leukocytes can promote further platelet recruitment to the liver [55].

## 4. The Role of Platelets in HSC Activation

During liver damage, quiescent vitamin-A storing HSC transition to an activated myofibroblast-like phenotype [43]. This activated state is regulated by a number of growth factors, cytokines, chemokines, and oxidative stress products, and is characterized by an increased production of ECM and morphological changes [56]. The excessive deposition of ECM results in severe liver stiffness and is a critical obstacle for anticancer drug infiltration in solid tumors [57,58,59]. In addition, these activated stellate cells are known to directly influence tumor cell growth and induce a more aggressive tumor phenotype [60].

Platelet-derived growth factor beta is a key element in activated stellate cell signaling and fibrinogenesis, as it strongly stimulates stellate cell activation and subsequent ECM deposition [61]. In physiologic conditions, its main source are platelet α-granules [27,62]. The pro-fibrotic PDGFβ signaling has a synergistic effect with TGFβ, another important driver of HSC activation and poor prognostic marker for HCC and liver fibrosis [12,62]. Tumor growth factor beta was shown to act as a central player in pro-tumorigenic HSC signaling and tumor development from early cancer stages to metastasis [58]. Several studies have examined the interactions between HSC and platelets, as well as the role of platelet-derived PDGFβ and TGFβ in chronic liver disease and HCC. A study on a biliary fibrosis mouse model revealed a significant increase in hepatic and serum levels of PDGFβ in the diseased mice, as well as a significant accumulation of platelets in the fibrotic liver compared to healthy controls [12]. This is in line with clinical findings showing that PDGF is increased in cirrhotic patients [63]. The in vivo study using MDR2-null mice which develop spontaneous biliary fibrosis, showed that platelet depletion brings hepatic levels of PDGFβ back to baseline levels and suppresses whole-liver expression of pro-fibrotic genes [12]. This indicates that platelet-produced PDGFβ is a key driver of HSC activation in vivo. In addition, PDGFβ has been reported as the most potent mitogen for cultured HSC isolated from rat, mouse or human liver and a key promoter of ECM component synthesis and fibrosis in vivo [57,64]. A study on transgenic mice over-expressing PDGF found that PDGF plays a central role in diverse stages of fibrosis and HCC, as PDGF-transgenic mice developed significantly more malignant transformations than controls. A suggested mechanism behind this PDGFβ-induced effect involves the upregulation of the TGFβ receptor [65]. Another study examined TGFβ-induced HSC activation in a hepatic fibrosis mouse model deficient in platelet-derived TGFβ. Results showed that these mice were partially protected from developing fibrosis and had less activated HSC compared to controls, suggesting that platelet-derived TGFβ is crucial for initiating pro-fibrotic HSC signaling [66]. A recent study on a CCl_4_-induced rat model for liver cirrhosis showed that the oral administration of anticoagulants, clopidogrel, and dabigatran, decreases the expression of TGFβ, smooth muscle actin, and collagen [67]. Rats treated with dabigatran and clopidogrel exhibited normalized biochemical and pathological changes, thus supporting the hypothesis that anticoagulant drugs may exert an anti-fibrotic effect.

Serotonin is another important regulator of stellate cell activation and it is stored at very high concentrations by platelets and released upon activation [38,64]. HSC strongly upregulate the expression of serotonin receptors-2A and -2B when they are activated. Signaling via serotonin receptors positively regulates the expression of TGFβ, collagen, and other pro-fibrotic factors, thus indicating an important role in fibrosis [64]. Studies have found that platelet-derived serotonin stimulates transdifferentiation of rat cardiac fibroblasts into myofibroblasts and enhances their migration and promotes TGFβ expression in rat HSC in vitro [68]. Targeting the serotonin-2A receptor with the inhibitor ketanserin alleviated biliary fibrosis in rats through blocking TGFβ signaling [69]. One of the most abundantly present chemokines in platelet α-granules is chemokine (C-X-C motif) ligand 4 (CXCL4), which was shown to mediate liver fibrosis in vitro and in vivo by promoting HSC proliferation, migration, and signaling and by playing an active role in the intrahepatic inflammatory response. Serum levels of CXCL4 were found to be elevated in patients with viral hepatitis and fibrosis, while the expression of CXCL4 in the liver was downregulated by the platelet inhibitor aspirin in a rat fibrosis model [52,70]. Kondo et al. showed that platelet accumulation in thrombocytopenic patients with chronic hepatitis C (HCV) was present in cirrhotic non-cancerous livers, which correlated with inflammation and activated HSC [71]. This finding suggests that platelet accumulation in the liver is involved in thrombocytopenia and liver fibrosis in HCV. Although these platelet-mediated effects on stellate cell activation were mainly seen on fibrosis, they might also be extrapolated to HCC, considering the importance of activated stellate cells in the onset and progression of liver cancer.

## 5. Platelets and the Hepatic Immune Response

Platelets have been increasingly recognized as potent drivers of both the innate and adaptive immune responses. Upon activation, platelets express adhesive and immune receptors and secrete a variety of mediators that modulate the inflammatory response and recruit immune cells [9]. As they do not leave the circulation, they mainly interact with immune cells in the liver and spleen [13]. By altering the immune-environment, platelets could contribute to the progression of HCC, as it has been reported that the upregulation of several pro-inflammatory genes (TNFα, IL-6 and CCL2), as well as NK- and T-cell infiltration are associated with longer survival [72]. One of the key factors mediating the inflammatory effect of platelets is serotonin. Platelets are key distributors of serotonin, which is a potent mediator in both innate and adaptive immunity, as platelets can ensure its targeted release in the right environmental setting, such as an inflammatory response.

Tumor-associated macrophages release growth factors which promote tumor cell proliferation. They also induce an “angiogenic switch” by secreting pro-angiogenic factors, influence HSC activation, and deepen the immunosuppressive phenotype of the inflammatory cell population, which further contributes to tumor progression and metastasis [43,73]. Platelet-derived microparticles have an immunomodulating effect on macrophages, which are attracted to the site of injury by chemotactic factors such as CXCL4, TGFβ, and PDGFβ. By downregulating mRNA expression and release of TNFα, CCL4, and CSF1 and enhancing their phagocytic capacity, platelet-derived microparticles have been shown to contribute to macrophage polarization towards a pro-tumoral phenotype [74,75,76], which is associated with poor survival in HCC [77]. A recent study in Nature Medicine by Malehmir et al, shows that Kupffer cells themselves are key players in recruiting intrahepatic platelets in a mouse model for non-alcoholic liver disease [78]. They further explain that platelet activation correlates with an increase in immune cell-attracting chemokines and cytokines and that platelet adhesion and activation—but not aggregation—are essential for non-alcoholic steatohepatitis and HCC. In addition, they present anti-platelet drugs as a therapeutic strategy for non-alcoholic liver disease and subsequent HCC development. It has also been noted in other studies that, while platelets need functional Kupffer cells in order to mediate tissue repair in liver injury, the absence of platelets results in a Kupffer cell phenotype that is much less harmful [79]. Platelet-derived serotonin has been consistently reported to inhibit TNFα production in stimulated monocytes and prime macrophages for anti-inflammatory signaling [38,80]. Serotonin has also been shown to upregulate a pro-fibrotic gene profile in macrophages through serotonin receptor 7-initiated PKA-dependent signaling [80]. In addition, studies have shown that human platelets suppress the macrophage’s anti-tumoral killing capacity by downregulating TNFα production in vitro [29]. These findings suggest that platelets play an important role in mediating the macrophage´s immune-response and could contribute to the occurrence of tumor-associated macrophages in the tumor micro-environment.

Platelets have also been shown to exert an immunosuppressive effect through direct contact with NK-cells and T-cells, as well as through paracrine signaling via TGFβ to reduce NK-cell anti-tumor activity [81]. In vivo and in vitro studies in other non-HCC tumor models have shown that major-histocompability complex-1 (MHC-1) can be transferred from platelets to tumor cells, allowing them to avoid recognition by NK-cells and impairing cytotoxicity [81]. Another aspect of the platelet-mediated hepatic immune response is the effect platelets exert on the infiltrating T-cell population in the liver. In the case of HCC in a background of chronic hepatitis B, platelets have been found to strongly promote the accumulation of functionally inefficient virus-specific CD8^+^ T-cells in vivo [82]. The virus-specific T-cell response was shown to maintain hepatocellular injury, while anti-platelet therapy led to a reduction in T-cell accumulation, which suppressed hepatocarcinogenesis and improved survival [13,14,83]. Reduced liver cell damage was also reported in virus-infected mice lacking platelet-derived serotonin, while serotonin treatment accelerated active CD8^+^ T-cell infiltration and promoted hepatitis progression [84].

## 6. Thrombocytosis and Thrombocytopenia in Liver Disease

Thrombocytosis has long been associated with various cancer types and has been reported to correlate with larger tumor sizes and shorter survival at the time of diagnosis [85,86]. However, reports on the association between platelet count and HCC progression have been widely inconsistent, as thrombocytosis and thrombocytopenia are both identified as risk factors in HCC development and prognosis [87,88]. This is probably due to the unique biological and clinical context of the cirrhotic liver that precedes most cases of HCC. Multiple changes occur in the hemostatic system as a result of decreased liver function, while the tumor itself further perturbs the hemostatic balance. One of the main determining factors of platelet count is thrombopoietin (TPO). Thrombopoietin is expressed both by normal and cancerous hepatocytes and regulates platelet mobilization and production in the bone marrow [89]. Studies have found that HCC patients with thrombocytosis have significantly higher mean serum levels of TPO and that gene expression of TPO mRNA is higher in the tumor tissue, compared to the surrounding non-tumoral tissue [87]. It is thus not surprising that higher platelet counts can be found in HCC patients with cirrhosis, compared to HCC-free cirrhotic controls [90].

Thrombocytosis-associated HCC often develops in well-compensated patients (Child A class) and is correlated with HCC metastasis along with an increase in blood AFP, higher portal vein thrombosis, large tumor size, and low cirrhosis [31,87,88,91]. However, a cohort study of 634 HCC patients revealed thrombocytosis association with large size tumors and improved liver functions [92]. Platelet-to-lymphocyte ratio has been proposed as an independent prognostic marker for patients with advanced HCC not receiving systemic sorafenib, as a study showed that a higher platelet-to-lymphocyte ratio was associated with shorter survival [93]. Mean platelet volume, a parameter of platelet size, has also been positively correlated with HCC progression [94]. Recent studies have suggested a post-operative platelet-to-lymphocyte and neutrophil-to-lymphocyte ratio as a combined tool to predict HCC recurrence and overall survival after surgical liver resection [95].

In contrast, thrombocytopenia-associated HCC is often related to small-size tumors, low blood albumin, and having a fibrotic/cirrhotic background [96]. Possible explanations for a lower platelet count in patients with liver disease are impaired TPO production by the liver, a reduced platelet half-life due to auto-antibodies and the fact that up to 90% of platelets can become sequestrated in the spleen during advanced liver cirrhosis [79]. Since it is mostly associated with liver fibrosis, a decrease in platelet count is correlated with increased pathological fibrosis scores with HCV-HCC patients [97]. Additionally, in a mouse model for hepatitis B (HBV), thrombocytopenia was associated with better disease outcome [83]. Therefore, thrombocytopenia is considered a predictive and prognostic factor in HCC. Recently, the united states food and drug administration (FDA) has approved Doptelet (Avatrombopag) to treat low platelet count in patients with chronic liver diseases, mainly due to the increased risk of bleeding associated with thrombocytopenia [98].

## 7. Targeting Platelets as a Therapeutic Strategy in Liver Disease

The platelet plasma membrane harbors several receptors that regulate platelet activation and aggregation, which causes the release of growth factors, cytokines, and chemokines that influence the different cells in the tumor and stromal compartment (Figure 2). Because of the platelet´s key role in modulating tumor cell behavior and their interaction with several cell types in the tumor microenvironment, they form an interesting therapeutic target for HCC.

Preclinical and clinical studies have demonstrated the protective effect of anti-platelet agents against HCC and other types of cancer. The most studied platelet activation inhibitors are aspirin and clopidogrel. Aspirin irreversibly inhibits cyclooxygenase-1 on platelets, which is involved in the metabolism of arachidonic acid through TXA2 synthase activity (Figure 2). Clopidogrel and ticagrelor are P2Y_12_ (purinergic, P2 receptor) inhibitors, which restrain binding of ADP to its receptor and attenuate the activation and aggregation of platelets (Figure 2). Several studies have shown that clopidogrel [102] and ticagrelor [78] inhibit the progression of HCC in vivo. Clinically, clopidogrel has been shown to reduce the risk of HCC in chronic HBV patients whose HBV is effectively suppressed [103]. However, anti-platelet therapy containing clopidogrel may increase the risk of bleeding, which has not been noted in treatment with aspirin [103]. In recent years, aspirin has generated significant interest as a potential chemopreventive agent. The largest breakthrough was the finding that a low-dose of aspirin decreases the risk of developing colorectal cancer [99] and could even decrease the progression of an established tumor [104]. Similar findings have been made in HCC, as clinical studies showed that long-term aspirin usage is associated with a dose-dependent reduction in HCC-risk [100,103,105,106]. It has also been shown that aspirin reduces the risk of liver fibrosis in patients who have been transplanted for hepatitis C [107] and is associated with an improved liver function and survival after chemo-embolization [108,109]. Interestingly, the effect of aspirin is not limited to its anticoagulant function, as aspirin can also directly decrease tumor cell proliferation [110], increase sensitivity to chemotherapeutics [101,111,112], and induce apoptosis [113]. As clopidogrel and aspirin bind to different platelet receptors—respectively P2Y_12_ and cyclooxygenase-1 (Figure 2)—the combination of aspirin and clopidogrel can work synergistically and improve the course of HCC progression through distinct pharmacological effects. The combination of aspirin and clopidogrel demonstrated inhibition of liver injury and attenuation of HCC-development in an HBV mouse model [14,15], and clinically reduced the risk of HCC in HBV patients [103]. A clinical study following outcomes for patients with HBV-related HCC after liver resection, showed that anti-platelet therapy with aspirin or clopidogrel significantly improved recurrence-free and overall survival after five years [114]. The anti-tumoral effect of aspirin and clopidogrel was also confirmed in a recent study, where this combination prevented non-alcoholic steatohepatitis and subsequent HCC development in several dietary and genetic mouse models [78]. This study also found that the P2Y_12_-inhibitor ticagrelor had a similar effect, while non-steroidal anti-inflammatory drugs did not affect the development of non-alcoholic steatohepatitis and HCC.

The inhibition of platelets can also be achieved by blocking of intracellular signaling pathways. Platelets possess three different isomers of phosphodiesterases (PDE2, PDE3, and PDE5) with different cyclic adenosine 3′,5′-monophosphate (cAMP) and cyclic guanosine 3′,5′-monophosphate (cGMP) selectivity (Figure 2). Both cAMP and cGMP are critical intracellular secondary messengers that exert a strong inhibitory effect on platelets [115]. Phosphodiesterases catalyze hydrolysis of cAMP and cGMP and limit the cyclic nucleotide intracellular levels, thereby regulating platelet activation. Reports have shown that PDE impairs cAMP and/or cGMP generation in various types of cancer and that the selective inhibition of PDE isoforms raises the levels of intracellular cAMP and/or cGMP, which induces apoptosis and cell cycle arrest [116]. A study suggested that PDE inhibitors stimulate liver regeneration after hepatectomy by preventing platelet aggregation and exerting an anti-inflammatory response [117]. Another important anti-platelet target is thromboxane 2 (TXA2), which is an active metabolite of arachidonic acid. Thromboxane 2 is a positive feedback lipid mediator and is synthesized by sequential oxygenation of arachidonic acid. It stimulates platelet aggregation, vasoconstriction and is involved in various pathological conditions [118]. Mechanistically, TXA2 induces platelet activation by binding to the thromboxane receptor on platelets (Figure 2). Thromboxane 2 synthase inhibitors prevent the conversion of prostaglandin-H2 to TXA2, thereby reducing TXA2 synthesis in platelets, whereas thromboxane receptor antagonists block the downstream effects of TXA2-receptors activation [118]. In a study on colon cancer, sodium ozagrel (TXA2 synthase inhibitor) has been shown to reduce hepatic metastasis in mice injected with colon cancer cells [119]. Thromboxane inhibitors have been found to attenuate fibrotic changes in a rat model for alcoholic cirrhosis [120], however not much is known about the effect on HCC.

The above evidence suggests the potential use of platelet-targeted pharmacological approaches to prevent liver cancer development and dissemination. More clinical studies and study cohorts with follow-ups on HCC-patients treated with anti-platelet drugs are needed to broaden the understanding of the underlying mechanisms of platelet-mediated liver disease and the potential of inhibiting platelet activation as a therapeutic option.

## 8. Conclusions

Hepatocellular carcinoma is an inflammatory-related cancer that usually occurs in the context of hepatic injury and inflammation. The pathogenesis of HCC is characterized by highly complex and intertwined multifactorial signaling pathways, which often occur as a result of an interplay between tumor cells and cells of the stroma. The tumor stroma is not just a passive bystander in the pathogenesis, but actively fuels tumor progression and modulates the environment so that it sustains tumor cell proliferation and metastasis. This stromal environment includes hepatic stellate cells, macrophages, endothelial cells, extracellular matrix proteins, and a variety of immune cells. During liver fibrosis and HCC, there is an increase of angiogenesis, which leads to the formation of leaky and dysfunctional blood vessels. These leaky blood vessels facilitate the interaction between platelets with different cells of the tumor and stromal compartments.

A growing body of evidence highlights platelets as potent mediators in HCC and other chronic liver diseases. Their involvement in pro-fibrinogenic signaling, the hepatic immune response and HCC proliferation and metastasis has been widely reported by both in vitro and in vivo studies. Anti-platelet therapy has been shown to ameliorate liver injury and improve the disease outcome in several in vivo models for chronic liver disease and HCC. Despite the increasing number of in vivo and in vitro reports on platelet involvement in hepatic pathophysiology, clinical evidence of their therapeutic benefit for HCC-patients remains scarce. In addition, the association between the blood platelet count and disease outcome is a controversial topic, as thrombocytosis and thrombocytopenia are both identified as risk factors in HCC development and prognosis. Likely the unique biological and clinical context of the cirrhotic liver preceding most cases of HCC causes these contradictory clinical findings. Therefore, further research is necessary to elucidate the role of platelets in the cirrhotic liver and to verify the therapeutic potential of anti-coagulant therapy for patients with HCC in a background of chronic liver disease.

## Figures and Tables

**Figure 1 cancers-11-01022-f001:**
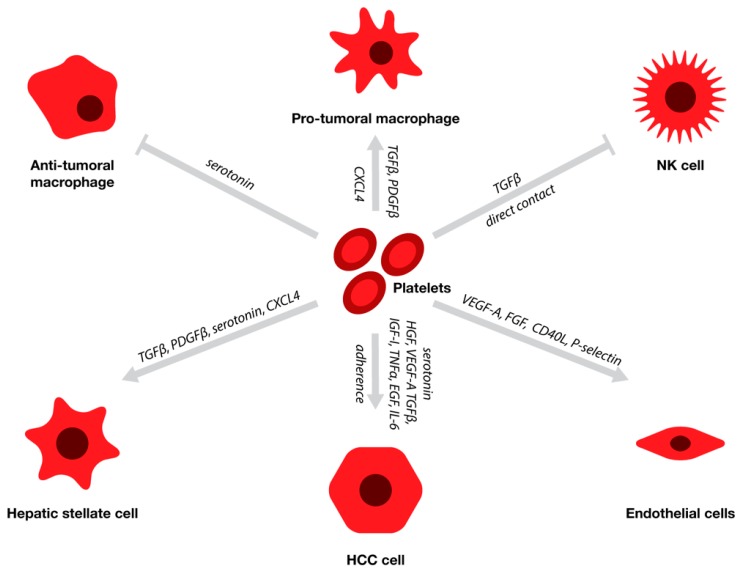
The interaction between platelets and different cells in the tumor and stromal compartments of hepatocellular carcinoma (HCC). Upon activation, platelets release α-granules and dense granules containing inflammatory cytokines, chemokines, and growth factors, such as platelet-derived growth factor beta (PDGFβ), serotonin, endothelial growth factor (EGF), insulin-like growth factor I (IGF-1) transforming growth factor beta (TGFβ), tumor necrosis factor alpha (TNFα), interleukin-6 (IL-6), chemokine (C-X-C motif) ligand 4 (CXCL4), vascular endothelial growth factor A (VEGF-A), hepatocyte growth factor (HGF), and fibroblast growth factor (FGF). Platelets promote HCC cell proliferation, invasion, and chemoresistance, by releasing cytokines (serotonin, HGF, VEGF-A TGFβ, IGF-1, TNFα, EGF, and IL-6) and by adhering to tumor cells, allowing them to avoid immunodetection. Their involvement goes beyond the direct effect on tumor cells, as they also affect the different cells in the stromal compartment, which creates an environment that stimulates tumor growth, invasion, and metastasis. Platelets release factors (mainly TGFβ, PDGFβ, serotonin, and CXCL4) that activate stellate cells, turning them in extracellular matrix (ECM) producing myofibroblasts which support tumor growth and aid metastasis. By secreting pro-angiogenic factors (VEGF-A and FGF) and by interacting with endothelial cells via CD40L and P-selectin, they contribute to the angiogenic switch, which increases the tumoral blood supply and facilitates vascular invasion. Platelets also play an important role in changing the hepatic immune cell population, as the release of microparticles containing serotonin, CXCL4, TGFβ, and PDGFβ promotes a shift from anti-tumoral macrophages to pro-tumoral macrophages, which are potent drivers of carcinogenesis. In addition, both the direct contact and release of TGFβ can decrease the cytotoxic potential of natural killer (NK) cells, causing an immunosuppressive environment that benefits tumor growth.

**Figure 2 cancers-11-01022-f002:**
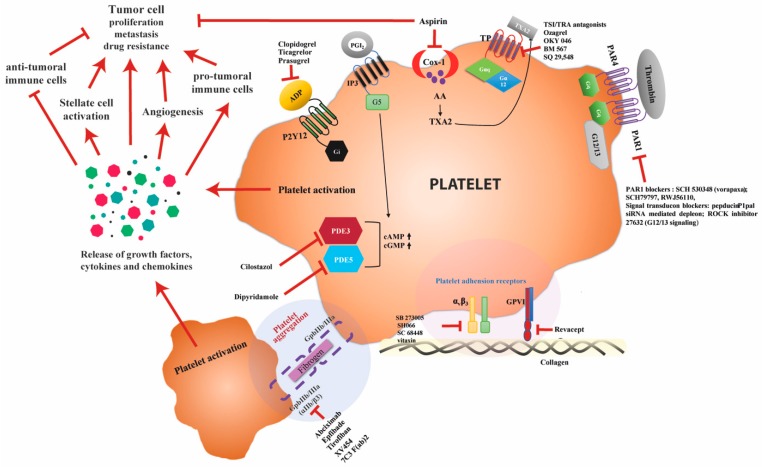
Overview of different anti-platelet therapies currently tested in preclinical and clinical studies for hepatocellular carcinoma (HCC) and other types of cancer. Aspirin irreversibly inhibits cyclooxygenase-1 on platelets, which is involved in the metabolism of arachidonic acid through TXA2 synthase activity. It has generated significant interest as a potential chemopreventive agent, mainly due to the finding that a low-dose of aspirin decreases the risk of colorectal cancer [99] and HCC [100]. Clopidogrel, on the other hand, is a purinergic P2Y_12_ receptor inhibitor. Clopidogrel restrains the binding of adenosine diphosphate (ADP) to its receptor and attenuates the activation and aggregation of platelets. Combining aspirin and clopidogrel has been shown to ameliorate the progression of HCC in vivo [14,15,78]. The effect is synergistic, through their distinct binding sites on platelets. Inhibition of platelets can also be achieved by targeting isomers of phosphodiesterases (PDE2, PDE3, and PDE5) which have different cyclic adenosine 3′,5′-monophosphate (cAMP) and cyclic guanosine 3′,5′-monophosphate (cGMP) selectivity. These drugs have been shown to induce apoptosis and cell cycle arrest in a broad spectrum of tumor cells and are known to stimulate liver regeneration after hepatectomy [101].

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
