# Peer review of "Platelets as Key Factors in Hepatocellular Carcinoma"

_cancers, 2019, doi:10.3390/cancers11071022_

Round 1

Reviewer 1 Report

Comment: (456 words)

Manuscript ID: cancers-540782, entitled: Platelets as key factors in hepatocellular carcinoma Hepatocellular Cancer Treatment (Authors: Natasa Pavlovic, Bhavna Rani, Pär Gerwins, Femke Heindryckx), attempted to update the field through their perspective of platelets-mediated chronic inflammation specifically regulated by the hepatic microenvironment. They singled out that potential effect and therapeutic value of platelets in the disease progression, which is of great interest. They offer their insight onto the effects of platelets on the haemostatic microenvironment in liver cirrhosis and cancer, which branch out a new path to therapeutic mechanisms by which they can act on platelets to synthesize, express, and release many bioactive molecules whose functions go beyond mediating haemostasis, leading to HSC and HCC. The clarity should be enhanced by addressing seven specifics below.

Specific Comment:

Page 2, Lines 56-59: “Tumor cells are also known to activate the coagulation

cascade by secreting thrombin and tissue factor, which results in a meshwork of platelets and fibrin that shields the tumor cells, allowing them to escape immune-surveillance and successfully invade distant sites [15].” Can they be specific in the liver?

Page 2, Fig.1 “bidirectional signaling between these factors that results in creating a tumor growth” – some narratives should be provided with citations to illustrate how pro-tumor signaling (pro-macrophages) transitions to anti-tumor signaling. Pro-tumor macrophages to NK cells: either using the arrow or blunt end line (choose a symbol for illustration of inhibitory effect); do not use current dual functions of the symbol. The illustration is of confusing: HCC converts to endothelia? HCC converts to HSC? Vice versa? How did that occur? If they mean platelet-mediated activation of the endothelium or HSC, they need to clarify by changing the illustrative scheme.

Page 8, Fig. 2: How did that work? Single-agent? Dual agents? Multiple agents? How could they orchestrate multiple agents? Side-effects? Any specific to liver cancer?

Page 8, Lines 337-361: they need to mark some text messages (molecules) on Fig. 2, which regulate HSC or HCC.

Page 9, line 388: “multifactorial signaling pathways, a large portion of which are yet to be fully understood” - they highlighted platelets effects, but they need to speculate in depth on how to turn the multifactorial agents to the potent mediators in HCC and other chronic liver diseases, which is expected from the reader.

Page 9, Lines 396-7: “while the association between blood platelet count and disease outcome remains a controversial topic.” – Why? Clinic reports? What bench marks do they come up to count part such a scheme?

Grammar errors and choice of style should be observed with standard English. E.g., Page 2, Fig.1 “bidirectional signaling between these factors that results in creating a tumor growth” – “result” is preferred as with “that” upfront “factors” - style.

Reviewer 2 Report

The litterature on this field does not yet allow, in our mind, to plan a review manuscript. Although the hypothesys of a pathophisyologic role of platelets in the occurrence of HCC is surely fashinating and interesting, the studies published on the topic regard mostly research in vitro that necessarely need to be supported by in vivo studies.

Reviewer 3 Report

Recently, platelet has been found to play an active role in liver disease. From 2016 to 2019, there are some review articles discussing the relationship between platelet, cancer, liver fibrosis or chronic liver disease which highlights the importance of platelet in different aspects. In this review article, by understanding the mechanisms between platelets and HCC allows to provide a novel and valuable therapeutic approach in the future clinical treatment. I suggest that the author can address the following questions:

The authors made too much segmentation in each section (especially in section 2, 3 and 4), it is hard to read. I suggest the author can merge some paragraph or giving the subtitle might also be a solution for better organized the paragraph.    

In section 5, the author is discussing the regulations between platelets and the hepatic immune response. Recently, there are an important finding of this topic that had been published in Nature Medicine this year. The article entitled “Platelet GPIbα is a mediator and potential interventional target for NASH and subsequent liver cancer.” From this research, they had found that platelet-mediated inflammation will cause NASH and carcinogenesis. The author should also include this new information in this review article.

In the article, the author has reported the target of platelets as a therapeutic approach in liver disease. However, since the chemotherapy is the main approach for the treatment of advanced HCC patients in the clinic, the author should also collect the relative researches to describe nowadays findings in the relationship of platelets and chemotherapy. For example, in article “Antagonism of sorafenib and regorafenib actions by platelet factors in hepatocellular carcinoma cell lines”, they had found that platelet inhibits the cytotoxicity of the chemotherapy drugs, sorafenib and regorafenib, in HCC.

There are some small errors in the manuscript, please check again carefully. For example, in line 30, there are two dots at the end of the sentence.

Round 2

Reviewer 2 Report

Based also o other reviewers the study reads better and in this present from may be accepted for publication.